# Optimizing Retrieval-augmented Reader Models via Token Elimination

**Moshe Berchansky**♠♡     **Peter Izsak**♠     **Avi Caciularu**♡♣
**Ido Dagan**♡     **Moshe Wasserblat**♠

♠Intel Labs, Israel     ♡Bar-Ilan University, Israel     ♣Google Research
{moshe.berchansky,peter.izsak,moshe.wasserblat}@intel.com

avica@google.com, dagan@cs.biu.ac.il

## Abstract

Fusion-in-Decoder (FiD) is an effective retrieval-augmented language model applied across a variety of open-domain tasks, such as question answering, fact checking, etc. In FiD, supporting passages are first retrieved and then processed using a generative model (Reader), which can cause a significant bottleneck in decoding time, particularly with long outputs. In this work, we analyze the contribution and necessity of all the retrieved passages to the performance of reader models, and propose eliminating some of the retrieved information, at the token level, that might not contribute essential information to the answer generation process. We demonstrate that our method can reduce run-time by up to 62.2%, with only a 2% reduction in performance, and in some cases, even improve the performance results.[1]

## 1   Introduction

The task of Open-Domain Question Answering (ODQA) (Voorhees, 1999) consists of answering questions using external knowledge, which is used as a source of relevant information that might be helpful for a *model* to extract or generate the right answer for a question. The expected answer can be short and concise (Kwiatkowski et al., 2019), or long and detailed (Fan et al., 2019), in which it is called Long-Form Question Answering (LFQA).

The *retriever-reader* architecture has been widely-used and adopted for ODQA tasks (Chen et al., 2017). The *retriever* fetches the most relevant passages using the question as a query. Then, the *reader* extracts or generates an answer, using the question and the relevant passages. The explicit structure of the system, consisting of these two sub-modules, allows for a decoupled optimization of either the retrieving or the reading process. In this work, we exclusively focus on the optimization

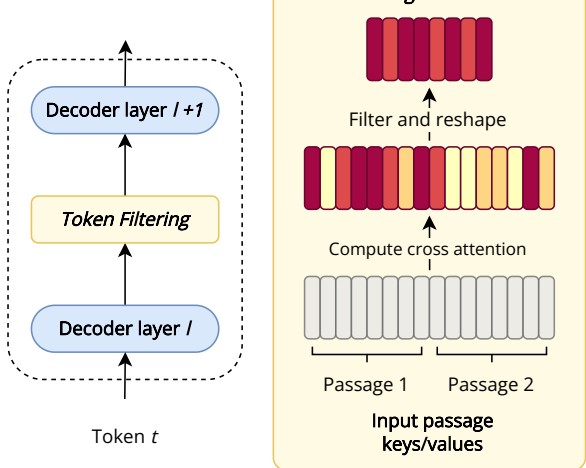

Figure 1: An overview of our Token Filtering method. Here, two passages are considered as the input to the decoder module. The Token Filtering operation is performed when generating token $t$, between decoder layer $l$ and $l+1$. Using the representation of token $t$ at layer $l$, the cross-attention is computed for all the *input* tokens. Then, we filter out the lowest ranked tokens from the input (marked in yellow), with only the highest ranking input tokens being used from the next generated token onward.

of the *reading* process. In order to assess ODQA methods, Petroni et al. (2021) presented a comprehensive evaluation framework that examines these methods in various open-domain tasks. Our study specifically concentrates on the Long-Form Question Answering task, utilizing the ELI5 dataset as a foundation (Fan et al., 2019).

There has been rapid and remarkable progress in retriever-reader systems for solving ODQA tasks using a generative approach (Sachan et al., 2021; Izacard et al., 2022). One such prominent approach is Fusion-in-Decoder (FiD) (Izacard and Grave, 2021b) that utilizes a generative text-to-text model to generate an answer. Despite the significant performance improvements, there are several computational bottlenecks associated with FiD that have

---

[1]We provide the source code for our work at https://github.com/mosheber/token_elimination.

a negative impact on efficiency. The most prominent ones are: (a) the need to retrieve a relatively large amount of documents to reach peak performance, and (b) an extensive cross-attention operation, caused by processing multiple concatenated retrieved passages, applied repeatedly for every generated token. These bottlenecks are negatively amplified in the case of Long-Form Question Answering.

Previous works have attempted to mitigate these bottlenecks, either by limiting the input to the reader or by directly optimizing it in a variety of methods. Yu et al. (2021) included a passage re-ranker inside the reader which aimed to filter out the least relevant passages during encoding. de Jong et al. (2022) optimized the decoder module by pretraining a modified and optimized architecture, and Ainslie et al. (2023) modified the attention operations performed to be less computationally intensive.

In this work, we tackle the heavy cross-attention computation in the decoder by introducing Token Filtering, a method that removes redundant tokens from input passages during the decoding stage, by dynamically computing their salience during generation. Using Token Filtering eliminates uninformative tokens from the cross-attention matrix, and prevents them from being utilized during answer generation, directly contributing to the reduction of the overall generation time. To further boost efficiency and reduce latency, we combine our Token Filtering approach with dynamic decoder layer skipping introduced by Schuster et al. (2022), referred to as CALM. By combining both approaches and by conducting experiments on three LFQA datasets, we find that this approach presents a better performance vs. efficiency trade-off than by using the methods separately, in most cases.

Overall, our contributions are as follows:

- We analyze the performance vs. efficiency trade-off of the FiD model, in terms of latency, FLOPs and the salience of the input information within the reader model, during long-form generation.

- We propose a novel approach for improving the efficiency of FiD, with a combined approach of Token Filtering and decoder layer reduction, which removes tokens and irrelevant layers during the generation process of every token for long-form answers.

- We show that models utilizing our approach can save up to 62.2% on the MS MARCO dataset, 54.9% on NQ, and 40.9% on ELI5, in terms of the generation time, while incurring a drop of no more than 2% in performance.

- Without computational restrictions, our method reaches state-of-the-art performance in KILT's ELI5 task.

## 2 Preliminaries

In a retriever-reader system, the reader, which is typically a language model, receives a query along with a collection of passages, where each passage often consists of a title and a context. Additionally, we are provided with the ground truth, which can be an expected answer or a gold passage that is most relevant to the query. Since our main focus is on generative models, we employ the widely-used Fusion-in-Decoder (FiD) model (Izacard and Grave, 2021b), which is a cutting-edge encoder-decoder model based on the T5 model (Raffel et al., 2020). The encoder module of the FiD model processes the input passages in parallel, with each layer taking the output of the previous layer, and the final output of the encoder is the output of its last layer. Similarly, each layer of the decoder processes its input by receiving the output of the preceding layer.

The decoder module then cross-attends to the large number of concatenated input representations and assimilates the information from the different passages to generate an answer. At each decoding step, the decoder computes the attention scores based on the precomputed input tokens' representations which serve as the query for the multi-headed attention operation, concurrently taking into account the current decoded sequence.

## 3 Efficiency Analysis

### 3.1 Encoder vs. Decoder Latency

There are multiple parts in a *retriever-reader* setup that have a direct effect on the end-to-end latency. One of them is potentially reducing the number of passages provided to the *reader* model. Izacard and Grave (2021c) evaluated the performance of FiD when decreasing the amount of passages provided to the reader, and found that the performance of the model drops as the number of input passages decreases. However, when excluding the bottom

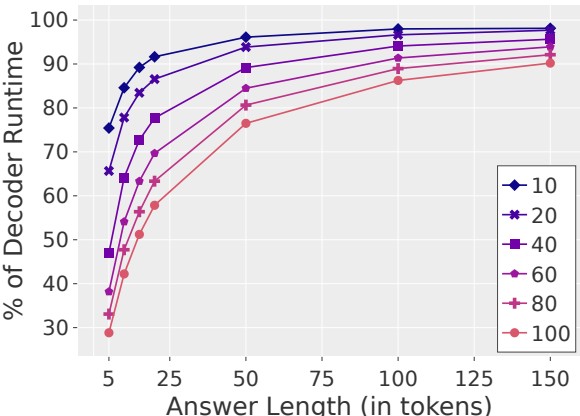

Figure 2: The percentage of the time (latency) of the decoder from the overall end-to-end latency, as a function of the number of generated tokens. Each color represents different amounts of input passages to the *reader*.

50% of the passages, the performance only drops by approximately 2%.

Naturally, the FiD latency could be reduced if we provide less input passages to the reader. However, it is unclear how much time is utilized by each of its sub-modules. de Jong et al. (2022) included a preliminary profiling of the execution time for the reader module of FiD. They show that even though the encoder is more expensive in terms of FLOPS computation, the decoder is more expensive in terms of actual latency.

Thus, we undertake an additional analysis, to comprehend how the time (latency) is distributed between the FiD encoder and the decoder modules, depending on the number of input passages and the amount of generated tokens. Our findings are illustrated in Figure 2. We artificially modify FiD to generate up to a fixed number of tokens. We observe that feeding a greater number of passages results in higher latency values for the encoder. However, as more output tokens are being generated, the share of the decoder of the total run-time significantly increases. Particularly, in the cases that the answer is long, we observe that regardless of the input number of passages provided to the reader, the majority of the time is spent in the decoder. Intriguingly, the ratio rapidly converges towards 100%, exceeding 50% after only 15 tokens.

Overall, in the specific case of long-answer tasks such as LFQA, we can conclude that the decoder serves as the primary source of latency and computational load during inference. This finding is further supported by similar works (de Jong et al., 2022; Hofstätter et al., 2022).

### 3.2 Cross-Attention Scores Analysis

An additional bottleneck affecting the efficiency of FiD is the extended sequence created by concatenating input passages, which the decoder focuses on during generation. Assuming the reader is supplied with an excessive amount of passages, our objective is to assess the importance of the input token representations. Essentially, our primary research question pertains to filtering out uninformative tokens that have no impact on answer generation, without compromising performance. Inspired by previous works that have assessed the relevance of input to decoders, we focus on the cross-attention scores. These scores have been recently demonstrated to serve as a metric of importance for the input token representations, particularly in relation to their impact on the accuracy of the answer (Caciularu et al., 2021; Izacard and Grave, 2021a; Izacard et al., 2022).

In order to investigate the utility of cross-attention scores as a meaningful indicator, we aim to verify their ability to focus on the important information within the input text. To accomplish this, we include the gold passage in a list of 100 retrieved passages (given a specific question). To simplify the analysis, we position the gold passage at rank 1, as the input matrix of the decoder's cross-attention matrix does not inherently incorporate any notion of order.

In order to examine the input token scores throughout the entire generation process, we calculate the average cross-attention scores for each decoder layer and at every generated token index. With the aim of identifying and filtering out irrelevant tokens, we select the top $p\%$ of tokens with the highest cross-attention scores and compute the proportion of the tokens that originate from the gold passage. Figure 3a demonstrates the outcomes of our investigation, where we selected $p = 10\%$ of the input tokens. This analysis was performed on a set of 1000 queries taken from the development set of ELI5, employing the FiD-Base model.

We observe that the decoder's initial layers (2nd and 3rd) exhibit the greatest proportion of tokens derived from the gold passage. This implies that these layers should be employed for calculating the input relevance scores. Additionally, we have noticed that, in most layers, the ratio reaches its peak around the 20th generated token and subsequently

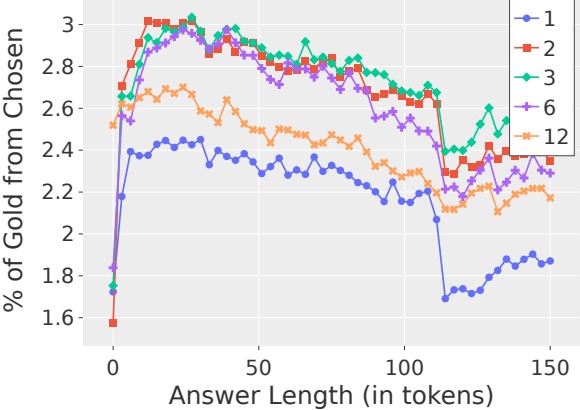 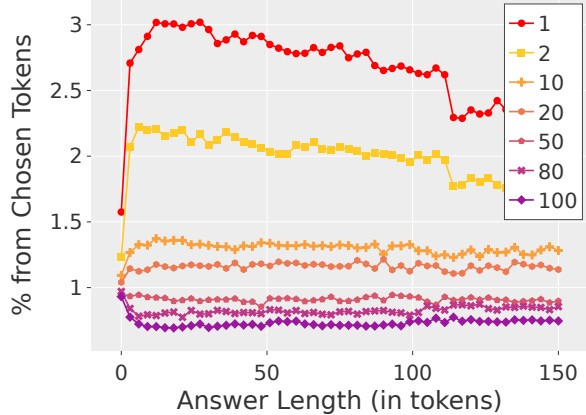

(a) The percentage of gold tokens for several chosen decoder layers.

(b) The distribution of passages over the chosen tokens, in the 2nd layer.

Figure 3: Cross attention score analysis when choosing $p = 10\%$ of the tokens, as a function of the generated answer length. **Left:** The ratio of tokens that were chosen from the gold passage, per decoder layer (1-12). **Right:** The percentage of tokens that were chosen from each passage (1-100). The gold passage (labeled as 1) is colored red.

declines during the generation process. This indicates that it is most advantageous to utilize the cross-attention scores in the early stages of generation.

Next, we proceed to examine the extent to which the model attends to tokens from the gold passage compared to other less informative tokens. The findings of this analysis are presented in Figure 3b, where we illustrate the number of selected tokens taken from each input passage specifically at the second layer. Our observations consistently indicate that the gold passage consistently maintains a notably higher proportion of selected tokens compared to any other passage, throughout the entirety of the generation process. Conversely, most of the passages exhibit ratios that are lower than what would be expected from a uniform distribution. Interestingly, we also note that the top passages exhibit higher ratios compared to the bottom ones.

In summary, we have demonstrated that the cross-attention scores possess the capability to prioritize the most pertinent information in the input, making them a reliable mechanism for selecting informative input tokens. Moreover, we have identified the optimal layers and ranges of generated token indices to generate these scores, ensuring the selection of the most appropriate input tokens. For a comprehensive examination of the cross-attention patterns, we encourage readers to refer to Appendix A for further details.

## 4 Method

Following our analysis in Section 3.2, we turn to implementing a method for filtering out the redundant information during the decoding stage. We aim to find a subset of the input tokens that is the most relevant for generating the correct answer. As pointed out previously, we can utilize the cross-attention scores computed between the generated tokens and the passages as basic signal for filtering out irrelevant tokens, similarly to Goyal et al. (2020).

Thus, we suggest a token filtering approach, using the cross-attention scores computed at a predetermined layer and generated token index during inference. At that point, for each input token, we compute the average attention scores over all attention heads, similarly to Caciularu et al. (2021); Izacard and Grave (2021a). Once these scores are computed, we keep the top k%-scored input tokens, which will be the only tokens to be considered towards the next tokens' predictions. Formally, the cross-attention scores per input token are defined as follows:

$$S_{t,l} = \frac{1}{h} \sum_{i=1}^{h} A_{t,l}^{i}, \quad (1)$$

where $t$ is the generated token index, $l$ is the layer index, $h$ is the number of attention heads, and $A_{t,l}^{i}$ represents the cross-attention scores at index $t$, layer $l$ and head $i$.

We perform a descending $argsort$ operation on

the scores above, and take the top $p\%$ from the sorted input token indices. Hence, we denote $T$ as the total number of input tokens from all the passages, and $T'$ as the amount of tokens we keep after filtering, which is $p\%$ from $T$:

$$Sorted_{t,l} = argsort(S_{t,l})$$
$$Top_{t,l} = (Sorted_{t,l}[i])_{i=1}^{T'}, \quad (2)$$

where $[i]$ indicates accessing the vector at index $i$. Finally, we keep only the tokens chosen in $Top_{t,l}$ from the cross-attention past key-values states $K_{past}, V_{past}$:

$$K_{past} = K_{past}[Top_{t,l}], V_{past} = V_{past}[Top_{t,l}], \quad (3)$$

where $A[B]$ selects the elements from $A$ whose indices appear in $B$. These new past key-value states are the only ones used for generating all subsequent tokens.

Since the filtering percentage, token index and layer can effect the quality of this mechanism, as inspected in Section 3.2, we obtain the optimal values for them by performing a hyperparameter-like search over their possible values, which is described in Section 5.4. We produce new past key and value representations for the input sequence (across all the decoder layers), containing only the selected tokens, resulting in a more compact tensor to attend to. We name the filtering mechanism Token Filtering, with an overview of the approach presented in Figure 1.

Note that we remove the irrelevant tokens from the keys and values of the encoder output sequence during inference time only once, hence reducing their dynamic dimension during computation for all the subsequent tokens. For additional details about the cross-attention scoring computation we refer the reader to Appendix B.

# 5 Experimental Setup

## 5.1 Datasets

Our experiments are conducted on commonly used datasets for LFQA.

**ELI5** (Fan et al., 2019) A dataset created from a Reddit forum named "Explain Like I'm Five". We use the train, validation and test sets as provided by the KILT benchmark[2](Petroni et al., 2020).

**MS MARCO** (Campos et al., 2016) A collection of crowd sourced responses to Bing queries. We use the Passage Ranking track, which consists of human generated natural and complete answers.

**NaturalQuestions (NQ)** (Kwiatkowski et al., 2019) A large-scale dataset by Google designed for natural language understanding and question answering research, consisting of real user queries from Google Search paired with corresponding Wikipedia passages.

For all datasets, we use the validation as the test set and a subset of the training set for validation, as done by Lee et al. (2019). We note that ELI5 is part of the KILT Benchmark[3], and thus is additionally evaluated on a held-out test set. We use the gold passages as the answers in MS MARCO and NQ. For a full specification of the dataset sizes, we refer to Table 5 in the Appendix.

## 5.2 Baseline Readers

We specify the hyperparameters used for training on the various datasets in Table 6 in the Appendix.

**FiD** We base our models on the FiD generative reader (Izacard and Grave, 2021c), which uses pre-trained T5 models (Wolf et al., 2019). We used the official implementation[4] of FiD throughout all our experiments.

**CALM** While our Token Filtering approach primarily focuses on eliminating redundant input tokens, it does not decrease the number of decoder layers responsible for processing them. To tackle this concern, we also incorporate a recent effective early exiting method for the decoder module (Schuster et al., 2022), known as CALM. We thus implement CALM and compare it to our method (see Schuster et al. (2022) for more information on the training scheme employed to train the FiD model, including the confidence classifier stage). In addition to independently evaluating CALM, we combine it together with our Token Filtering approach, resulting in a combined approach referred to as **Combined**.

## 5.3 Implementation Details

**Retrieval** We first create an index for retrieval over a Wikipedia dump[5], comprised of multiple

---

[2]https://huggingface.co/datasets/kilt_tasks

[3]https://eval.ai/web/challenges/challenge-page/689/leaderboard/1908/ROUGE-L
[4]https://github.com/facebookresearch/FiD
[5]https://huggingface.co/datasets/kilt_wikipedia

passages. For all the evaluated datasets, we retrieve 100 passages for each question from the index, using a combination of dense and sparse passage rankers. We refer the reader to Appendix C for more details regarding the retrieval process.

**Hardware**  We used 8 24GB NVIDIA RTX3090 for training base-sized models, and 8 40GB A100 GPUs for training large-sized models. For inference and latency measurements we used a single accelerator.

**Inference setup**  Throughout our latency measurements, we used a batch size of 1 and averaged the latency over all queries. Decoding is done using beam-search with 4 beams, and similarly as (Su et al., 2022) we limit the generated answer length to 300 tokens. We also control the minimal answer length per dataset, which we specify in Table 4 in the Appendix.

### 5.4 Performance vs. Efficiency Evaluation Process

We use KILT's implementation of ROUGE-L and F1 for performance measurements[6]. We measure efficiency as end-to-end latency in seconds for generating an answer to a question. To evaluate each method, on each dataset, we focus on the performance vs. efficiency trade-off, particularly on ROUGE-L vs. latency. For the evaluation of a given method (for example Token Filtering), we perform a hyperparameter search over multiple combinations on the *development set*, and end up with a collection of 2-dimensional points, where the x-axis is the latency, and the y-axis is the performance (ROUGE-L). Each latency-performance measurement is averaged across all questions in the development set. Then, we take the maximum and minimum over the observed values of the x-axis, and divide the resulting range into equally-sized intervals. In our experiments, we use 30 distinct intervals. For each interval, we find it's representative point, by taking the point with the maximum y-axis value from all the combinations in interval. Once all such points are determined per interval, they form a curve, which we name as the Max Curve of the method. We visualize the process in Figure 4, where the line in blue is the Max Curve.

Thus, for a method to be better than another, the Max Curve for it should be above and to the left of the curve of the other, meaning that it reaches equivalent results for less resources. Using the

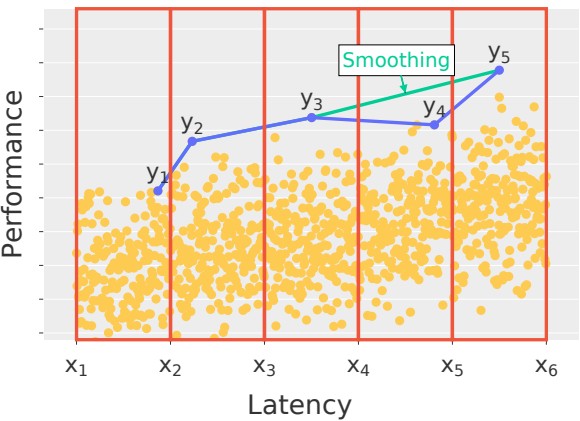

Figure 4: For Performance (y-axis) vs. Latency (x-axis), we divide the x-axis into 5 intervals over all hyperparameter combination results on the development set, represented as yellow dots. For each interval, we choose the combination with the best performance (i.e. y-axis value), thus forming the Max Curve in blue, with its smoothed version in green.

curve, we find the best hyperparameters for each method *per* interval, by selecting the hyperparameters of the representative point in the current interval. Finally, we take the best setting per interval, and run each one on the test set. For each of our methods, we produce a smoothed version, as the results are not necessarily monotonically increasing, which is shown in Figure 4 as the green line. These smoothed curves are the ones showcased in the final results in Figure 5.

When performing the search over the hyperparameter space, we used grid search, with the hyperparameters and their value ranges being specified in Table 7 in the Appendix. Other methods (such as random search, Bayesian Optimization (Snoek et al., 2012), etc.) may be attempted just as well in future works.

## 6  Experimental Results

### 6.1  Main Trade-off Comparison

In Figure 5, we showcase the performance vs. efficiency trade-off in terms of ROUGE-L and latency on the test set of each dataset. These results are the ones obtained after performing the hyperparameter optimization procedure stated described Section 5.4. The methods shown are the standard FiD model, CALM, Token Filtering, and Combined.

For the base-sized models (Figures 5a, 5b, 5c), we can observe all methods improve upon the baseline model, each one in a different aspect. For CALM, the method is able to reach lower latency

---
[6]https://github.com/facebookresearch/KILT

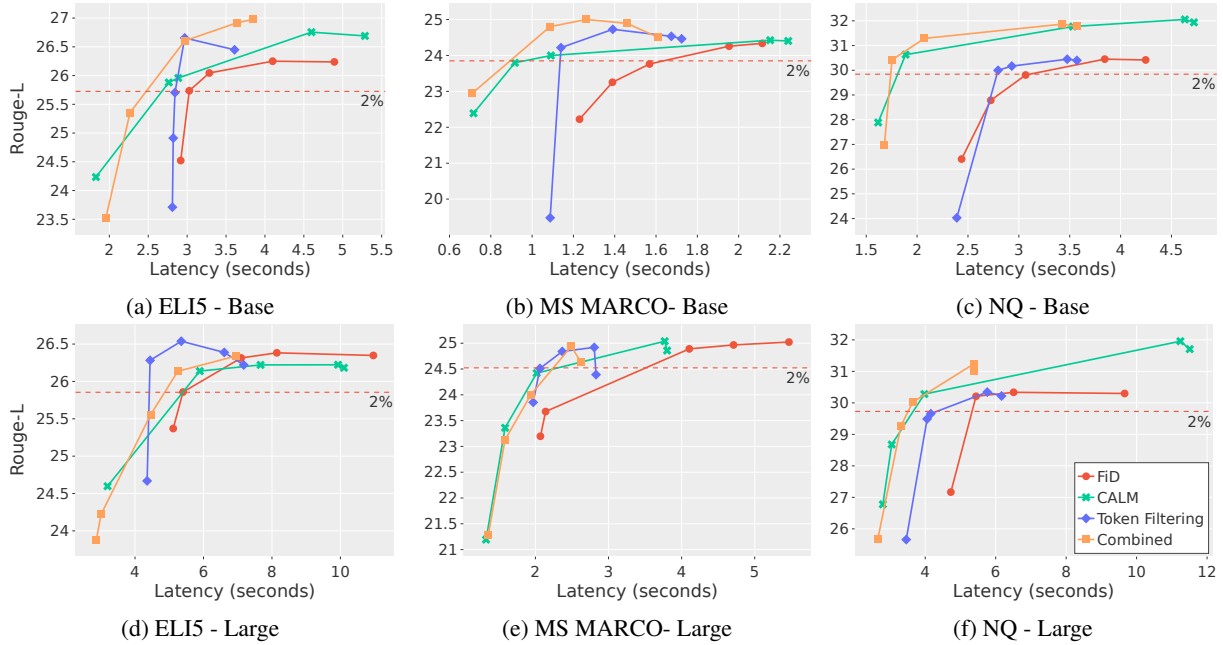

Figure 5: ROUGE-L Performance results of the different methods on the test sets, plotted as smoothed Max Curves, as a function of latency (seconds), for Base (top) and Large (bottom) models. Overall, our combined approach is able to reach a better trade-off than the regular FiD model, for most cases.

values, due to skipping the redundant layer computations. In the case of Token Filtering, it is also able to preserve and at times improve the performance of the model overall, while the latency improvement remains limited, since it is still computing the remaining tokens across all decoder layers. The performance improvement is presumably due to the redundant tokens being removed early on during the generation process, hence allowing the model to better attend to the salient information in the input.

When combining both methods, the performance enhancement of the Token Filtering and the latency reduction of CALM produce a better curve than either method alone. In addition, we showcase the drop in 2% performance per dataset, showing that our method is able to reduce the latency significantly more than the regular FiD, with the best reduction reached on the MS MARCO dataset for FiD-Base, saving 62.2% of the latency. In the NQ dataset however, for both the base-sized and large-sized models, while the CALM method does achieve proper latency reduction, the Token Filtering does not effect the results significantly. Since we focus on real-world scenarios, we showcase the trade-off with the **actual** latency, instead of measurements such as FLOPS (MACs), as done by previous works (de Jong et al., 2022). For those, we refer to Figure 6 in the Appendix for the trade-off

and FLOPS (MACs) analysis. For the large-sized models (Figures 5d, 5e, and 5f), we observe similar patterns to those in the base-sized variations, with the latency values being significantly larger. We note that the overall performance values for these models are not substantially different than those produced by the smaller versions, hence we do not focus as much on them.

## 6.2 Performance Comparison

To asses the performance of our Combined method further, we choose the best performing hyperparameter setting for the FiD-Base model, and report the test set results for each dataset, with Table 1 showing the results, compared to approaches suggested in Su et al. (2022). In particular, we compare to their implementation of FiD (named RBG FID) and their suggested system (named RBG), with the results of both taken from their published work. We note that both our models and the RBG models are of the same size. In our experiments, we denote the original FiD model we trained as FiD (ours), the FiD model with the Token Filtering method as FiD TF, and the Combined method as FiD Comb. On both datasets, FiD (ours), FiD TF and FiD Comb achieve state-of-the-art results, with Combined reaching the best overall performance in terms of ROUGE-L and F1 (aside from MS MARCO F1, where our approach is second).

| | ELI5 | | MS MARCO | |
| --- | --- | --- | --- | --- |
| | R-L | F1 | R-L | F1 |
| RBG FiD | 25.70 | 28.55 | 24.64 | 27.08 |
| RBG | 26.46 | 29.04 | 24.72 | **27.52** |
| FiD (ours) | 26.24 | 31.46 | 24.33 | 27.11 |
| FiD TF | 26.65 | 30.32 | 24.75 | 27.26 |
| FiD Comb | **26.97** | **31.76** | **25.11** | 27.41 |

Table 1: A comparison of the performance of our model, in comparison with the RBG model, where FiD TF stands for FiD with Token Filtering. Highest performance values are marked in **bold**, where R-L is ROUGE-L. We note that the results for RBG's models were taken directly from their published paper.

| Model | ELI5 | MS MARCO |
| --- | --- | --- |
| FiD (Ours) | 83.68 | 85.07 |
| FiD Comb | 83.79 | 85.27 |

Table 2: The BERTScore F1 results on the test set for both the original FiD model and our FiD Comb method, with each column indicating a different dataset.

As an additional point of comparison, we compute the BERTScore (Zhang et al., 2019) metric on the original FiD model, and our FiD Comb method, and present our findings in Table 2. We observe that our model performs on par and even a bit better overall than the original FiD implementation. We present a supplementary comparison of some of the answers generated by our method for the ELI5 test set in Table 8 in the Appendix. In addition, at the time of writing this paper, our Combined is ranked at the #1 position in terms of ROUGE-L and F1 on the ELI5 KILT leaderboard, with Table 3 containing all leaderboard results[7].

## 7 Related Work

**Open-Domain Question Answering**  Many previous works utilized setups which are based on the retriever and the language model reader components (Chen et al., 2017; Guu et al., 2020; Lewis et al., 2020). The goal of the retriever is to fetch the most relevant passages to a given question (Karpukhin et al., 2020). The reader processes the question and the relevant passages to extract or generate the answer, with generative approaches

---

[7]Since the results in Table 3 are on a hidden test set for the leaderboard, the results are different from those reported in Table 1.

| Model | ROUGE-L | F1 |
| --- | --- | --- |
| Krishna et al. (2021a) | 23.36 | 23.14 |
| RBG | 24.53 | 27.13 |
| FiD TF | 25.52 | 28.49 |
| FiD Comb | **25.61** | **29.99** |

Table 3: The published results on the official ELI5 test set, provided by the KILT leaderboard. FiD Comb runs with the same setting as best performing combination of the combined approach in Figure 5a.

achieve substantially better results (Izacard and Grave, 2021b). Subsequent works (Su et al., 2022; Krishna et al., 2021b) have adapted these generative approaches to produce long-form answers as well.

**Encoder-Decoder Efficiency**  Due to computation bottlenecks in generative models, particularly in encoder-decoder models (Vaswani et al., 2017; Raffel et al., 2020), previous works attempt to mitigate them. The encoder model can be used to filter out irrelevant passages during generation (Yu et al., 2021; de Jong et al., 2023). Nevertheless, the encoder's impact on the model's latency in long-form scenarios is negligible, as depicted in Figure 2. Consequently, our research centers on analyzing the computational aspects of the *decoder* instead.

In particular, the decoder model utilizes many redundant and heavy cross-attention operations, which can be removed or replaced with simpler alternatives (de Jong et al., 2022; Ainslie et al., 2023).

Since encoder-decoder models perform compute heavy operations in multiple layers, previous works have proposed stopping the layer propagation dynamically by assessing the model's confidence for prediction at a certain layer (Teerapittayanon et al., 2016; Schwartz et al., 2020). Other works have adapted this mechanism to decoder models as well (Schuster et al., 2022; Elbayad et al., 2019). However, these studies fail to tackle the issue of input size during generation, thereby resulting in computations being performed on irrelevant input to some degree, which we address through complementary token filtering.

**Data Reduction for an Efficient Computation** The input to encoder models tends to become increasingly large, especially in ODQA settings with many input passages. Since not all spans of infor-

mation are relevant to produce the correct answer, previous works propose eliminating the irrelevant tokens from the input to the encoder, by identifying the salient information during inference time (Goyal et al., 2020; Kim and Cho, 2020). Although these techniques effectively decrease computation at each layer, they are implemented in the encoder model rather than the decoder, which has been previously determined to have a more significant influence on latency. Our work leverages the cross-attention in the decoder early on during generation, thus effectively filtering the input tokens.

Qin and Durme (2023) suggested transforming language into a representation, by selecting a dynamically determined subset of input tokens, with these "nuggets" being acquired through tasks such as machine translation. However, our method doesn't incorporate any learning, focusing on analyzing the necessary input tokens for direct decoding instead.

Wingate et al. (2022); Mu et al. (2023) proposed prompt compression techniques for minimizing the amount of token vectors required to represent the same text. We note that our work does not discuss such aspects, given the context of questions and passages in our inputs.

## 8 Conclusions

We analyze the precise performance vs. efficiency trade-off of the FiD's encoder and decoder in long-form settings, with an analysis of the cross-attention operation in the *decoder* model. We show that the decoder has more impact on the latency, particularly for long outputs, and that the decoder attends to more salient information early on during generation. Hence, our proposed approach for efficiency reduction, namely a combined approach of Token Filtering and CALM, removes irrelevant layers and tokens during the *generation* process, for every token produced. Our approach achieves a significant reduction in resources (up to 62.2%), while not sacrificing more than 2% of the performance, and is the current state-of-the-art on the ELI5 KILT leaderboard. Future work can further develop a more dynamic method for choosing the most relevant tokens from the input, instead of using predetermined hyperparameters, and train the cross-attention patterns to better attend to the salient information during generation.

## Limitations

Regarding the retriever, as mentioned in Section 5.2, we did not experiment with a vast array of retrievers, due to the scope of the work being on the reader model.

Regarding the models for comparison, we primarily focused on the performance of the FiD model versus our own approach, while testing them on various datasets. Hence, we did not perform extensive reproductions of other methods, such other encoder-decoder models, but instead report their original results as they were published. We believe that our results can be generalized to other architectures as well.

In our hyperparameter search, we chose a subspace of all the possible values each parameter has, due to a limited amount of computation available. Our approximation of the space covers the main areas of relevance for our purposes.

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

# A  Cross Attention Pattern Analysis

In this section, we continue our discussion from section 3.2, regarding the analysis of the cross-attention scores.

In Figure 7, we present multiple versions of the plot in Figure 3a, with the rows indicating the dataset (ELI5, MS MARCO, NQ), and the columns representing a different percent of chosen tokens (10, 30, 50). For MS MARCO and NQ in 10%, the percentage of the gold passage tokens remains high for the lower layers, starting from token 10. The other layers do not reach the same percentage and degrade during the generation process. When increasing the percentage to 30, and later 50, the percentage of the gold passage is getting reduced substantially.

In Figure 8, we showcase an extended version of Figure 3b, for the various datasets and chosen token percentages as in Figure 7, with the rows and columns being similarly organized. For 10%, the gold passage gets the most tokens out of all the rest, for all datasets, with the lower passages getting less than 1%. However, for 30%, the gold passage is no longer the highest ranking for some of the datasets (MS MARCO, NQ), with the upper passages reaching higher, and the lower ones still being at the 1% mark. At 50%, the gold passage is no longer the most prominent, with it being nearly as insignificant as the lower passages in the case of MSMARCO. This suggests that increasing the percentage of tokens taken introduces unnecessary noise to the selected tokens, thus forcing the model to receive input from lower ranked passages. For MS MARCO and NQ in 10%, the percentage of the gold passage tokens remains high for the lower layers, starting from token 10. While the results above were done using an FiD-Base model, similar patterns are present for FiD-Large models through all previously discussed aspects.

# B  Attention Score Computation Extensions

In addition to the methods introduced in 4, the computation of the cross-attention scores can be further altered in a few key areas, which we tackle as well.

**Value Normalization.** As mentioned in Izacard et al. (2022), the scores can benefit from scaling by the $l_2$ normalized values tensor $V$. Thus, we can instead transform $A_{t,l}^i$ into:

$$A_{t,l}^i[n] = A_{t,l}^i[n]v_n \qquad (4)$$

,

where $[n]$ is the $n^{th}$ row, in this case the $n^{th}$ token, and $v_n = ||V[n]||_2$ is the norm of the $n^{th}$ row (token) in $V$. Hence, we apply this normalization to the attention scoring operation.

**Mean over all decoder layers.** Instead of taking the representation of the current decoder layer only, we instead take the average over every layer before the current one. Thus, we compute the attention scores $S_{t,l}$ for the input tokens as follows:

$$S_{t,l} = \frac{1}{lh} \sum_{l' \in [1,l]} \sum_{i \in [1,h]} A_{t,l'}^i \qquad (5)$$

From our preliminary analysis, this mean operation does not effect the quality of the filtering method, and hence is not applied.

## C   Retrieval Details

Since our method primarily focuses on the reader model, we have implemented a generalized approach for creating ranked passage lists. Our document corpus is taken from a Wikipedia Dump, which has been split into 100-word-long passages, as done in Karpukhin et al. (2020), including the article title. These documents are then stored in an Elasticsearch[8] index. Given a question from a dataset, we use BM25 over the passage index, to retrieve 250 documents. Then, we re-rank the passages using a sentence transformer[9](Reimers and Gurevych, 2019) model that was finetuned on the dataset, and keep only the top 100 ranked documents.

## D   Method Implementation Details

For the CALM, we utilize beam search for long sequence generation. In the beam-search setting, we use $n_b$ beams, there which causes the issue of how to allow some tokens to cease computation at a certain level, while allowing the others to continue computation. For the scope of our work, we apply the hard assumption that the confidence value is the lowest one from all beams, hence exiting only if all the tokens in the beams have satisfied the exiting condition. Formally, given confidence scores $\bar{c}_l = (c_l^1, c_l^2, ..., c_l^{n_b})$ at layer $l$, the confidence value used at the layer will thus be $c_l = \min_{j \in [1,n_b]} c_l^j$.

[8] https://www.elastic.co
[9] multi-qa-mpnet-base-dot-v1

We note that while Schuster et al. (2022) utilized a complex threshold calibration system, we instead showcase the effect of the various thresholding settings, once applied to the decoder model.

For the Token Filtering, since we are discussing mainly Encoder-Decoder architectures, we apply the filtering by removing the redundant tokens from the past key and value states for the cross-attention. In addition, we also remove said tokens from the encoder hidden states, encoder attention mask, and the encoder-decoder position bias.

| Dataset | Len. Chosen |
|---|---|
| ELI5 | 150 |
| MS MARCO | 50 |
| NQ | 50 |

Table 4: The chosen minimum answer length for during evaluation.

| Dataset | Train | Dev | Test | KILT |
|---|---|---|---|---|
| ELI5 | 272634 | 3000 | 1507 | 600 |
| MS MARCO | 498000 | 3000 | 6980 | - |
| NQ | 55622 | 3000 | 6489 | - |

Table 5: Sizes of the datasets, per train, dev and test respectively. We include the size of the KILT test set size, which we evaluate on separately.

| Parameter | Value |
|---|---|
| Base Model | T5-Base, T5 Large |
| Optimizer | AdamW |
| Max Seq. Length | 235 |
| LR | 5e-5 |
| LR Scheduler | Linear |
| Weight Decay | 0.01 |
| Precision | torch.bfloat16 |
| Batch Size | 64 |
| Training Steps | 60000 |
| Warmup Steps | 1000 |

Table 6: The training parameters used for training the FiD model on each dataset.

| Parameter | Ranges |
|---|---|
| General Parameters | |
| Input Psg. | [5, 100] |
| CALM | |
| Confidence Threshold | [0.2, 0.9] |
| Threshold Coef. | {0.5, 0.7, 0.9} |
| Threshold Decay | {3, 4, 5} |
| Token Filtering | |
| % of Input | {10, 30, 50} |
| Filtering Token | [1,20] |
| Filtering Layer | [1,$L$] |

Table 7: The hyperparameter search space for the CALM and Token Filtering methods, including parameters for all methods (Input Psg.), and the combined approach. {} indicate a set of possible values, while $[min, max]$ correspond to a range of values from $min$ to $max$. As in the paper, $L$ is the number of layers in the decoder.

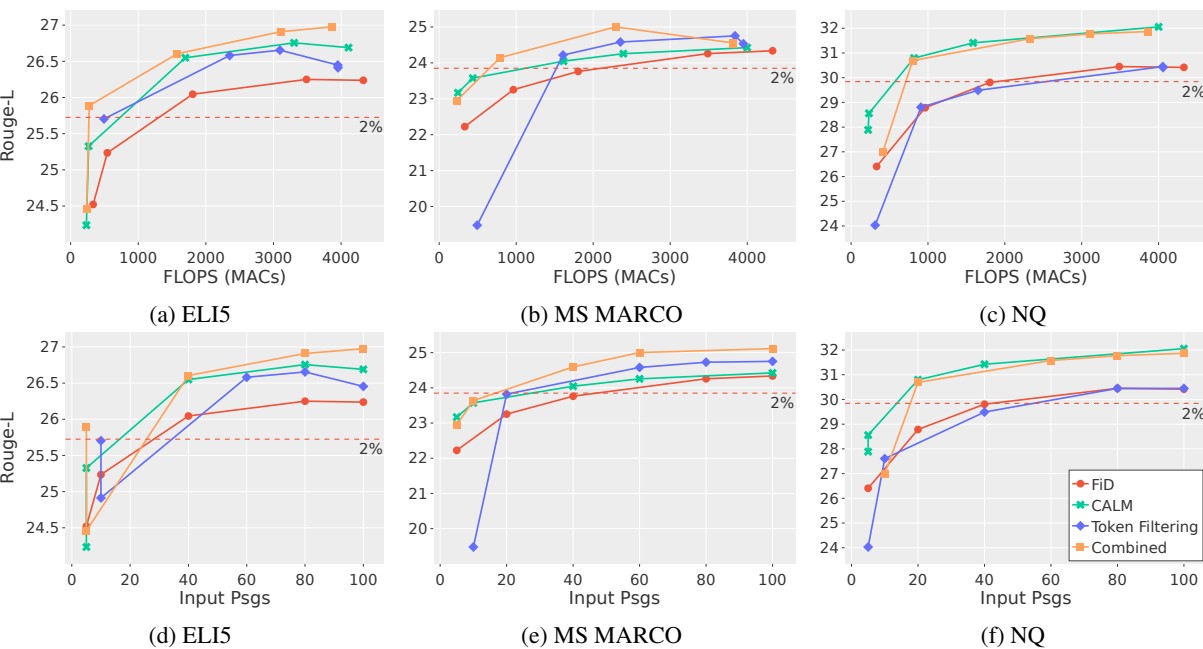

Figure 6: The ROUGE-L performance on FiD-Base vs. the FLOPS (MACs) (First row), and vs. the input passage amount (Input Psg., second row). The results are on the test set for each dataset, for the various methods utilized. We observe that the trends of the FLOPS (MACs) and the Input Psg. are very similar, since the passages effect the encoder the most, and the encoder has the most impact on FLOPS (MACs) (as stated by de Jong et al. (2022)).

| Question | Answers |
|---|---|
| How do you know something is done/said by Anonymous, when they are, by definition, anonymous? Can anyone do stuff/say things on behalf of Abobynous, or do they somehow have a recognized leadership who are not anonymous to each other/their members? | **Gold**: Any unknown person can do or say things in the name of Anonymous. That's why you basically have to ignore everything Anonymous says they will do, and only look at what they actually do. There's no leadership, but there are I believe certain core groups who know one another by persistent pseudonyms, and may possibly know one another afk. They are probably disproportionately influential, in that they run widely-subscribed youtube/twitter/whatever accounts which play a significant role in getting messages out to the rest of Anonymous, but they aren't leadership in any classical sense of the word. |
| | **FiD (ours)**: I'm not sure what you mean by "anonymous". Anonymous is a group of people who are working together to solve problems. They aren't anonymous to each other, they're just like any other group. There's no way to know if something is done by Anonymous or not, but you can be sure that it is. You can also be certain that someone is doing something that is not Anonymous, and that they are doing it for the benefit of others. If you've ever heard of someone saying something like "I am an Anonymous member, I am not an anonymous member" then you know that you are an "Anon" member. Then you have to be able to prove that the person who made the post was Anonymous. |
| | **FiD Comb**: I'm not sure what you mean by "anonymous" by definition. Anonymous is a loose hacktivist collective. They don't have an official leadership, but they do have some sort of "leadership" in place. _URL_0_ It's not that they're able to do anything, they just use the name "Anon" to refer to the members of the group, and they can be viewed as "a group of people who believe in the idea of anonymity" or something like that. There is no way to tell if someone is an anonymous group. It is possible for someone to be identified as an Anonymous member, or even have their own leader. |
| why is cranberry juice good for your kidneys? | **Gold**: It isn't, particularly. Some people use to think that the acid in cranberries stopped bacteria growing but thats been shown to be wrong. There is some vague evidence that cranberry juice can help *prevent* UTIs, though. Its thought that cranberries might contain a chemical that stops bacteria sticking to the cells lining the walls of the urinary tract, but no-one knows for sure. Once you have a UTI, though, no juice will cure it. The real reason that cranberry juice, lemon barley water etc are good for your kidneys is that you need to flush them out. Drink at least 8 glasses of water a day (2-3 litres of fluid daily), THAT is what is good for your kidneys. My daughter gets a LOT of UTIs, even with prophylactic antibiotics, and several members of my family have died from kidney disease. All I ever hear from well-intentioned friends is "have you tried cranberry and/or barley water?". Sigh. |
| | **FiD (ours)**: It isn't. There is no evidence that cranberry juice is good for your kidneys. It's just that there is not enough evidence to suggest that it is a good thing for you. _URL_0_ TL;DR - if you have an infection, you should not drink it. If you're on dialysis, it can lead to urinary tract infections, which can be fatal. The reason for this is because the bacteria that cause the infection can get stuck in your urine, causing the urine to become inflamed and irritated, leading to pyuria and kidney stones. This is why people who are in the throes of kidney problems should drink more than they normally do. |
| | **FiD Comb**: It isn't. There's no scientific evidence that cranberry juice is good for your kidneys. However, there are studies that suggest that it may be beneficial for you if you have a urinary tract infection. _URL_0_ TL;DR: Cranberries are incredibly acidic, so they can be sour and tart, which is why they're bad for the kidney. It also has some anti-coagulants in it that can help prevent the formation of bacterial plaques in the urine, reducing the amount of urine that is excreted from the bloodstream, and preventing the clotting of the bladder. This is also why some people who have kidney stones are more likely to get kidney stone formation. |

Table 8: ELI5 test set answers from the standard FiD model, FiD (ours), and our method (FiD Comb), with the Gold Answer as a reference.

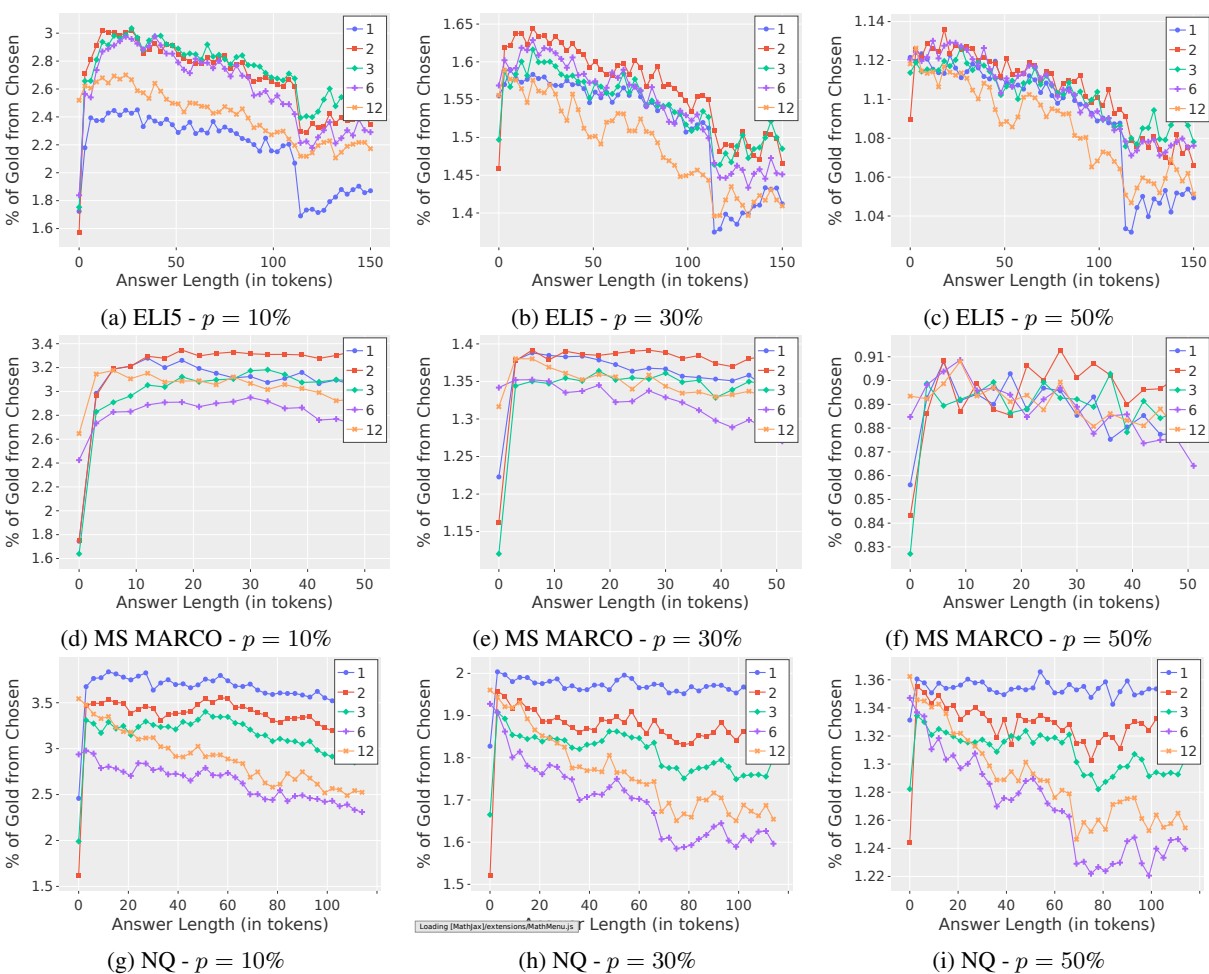

Figure 7: The ratio of tokens that were chosen from the gold passage, per decoder layer (1-12), for FiD-Base models. Each row represents a different dataset, and every column represents a different filtering percentage (10,30,50).

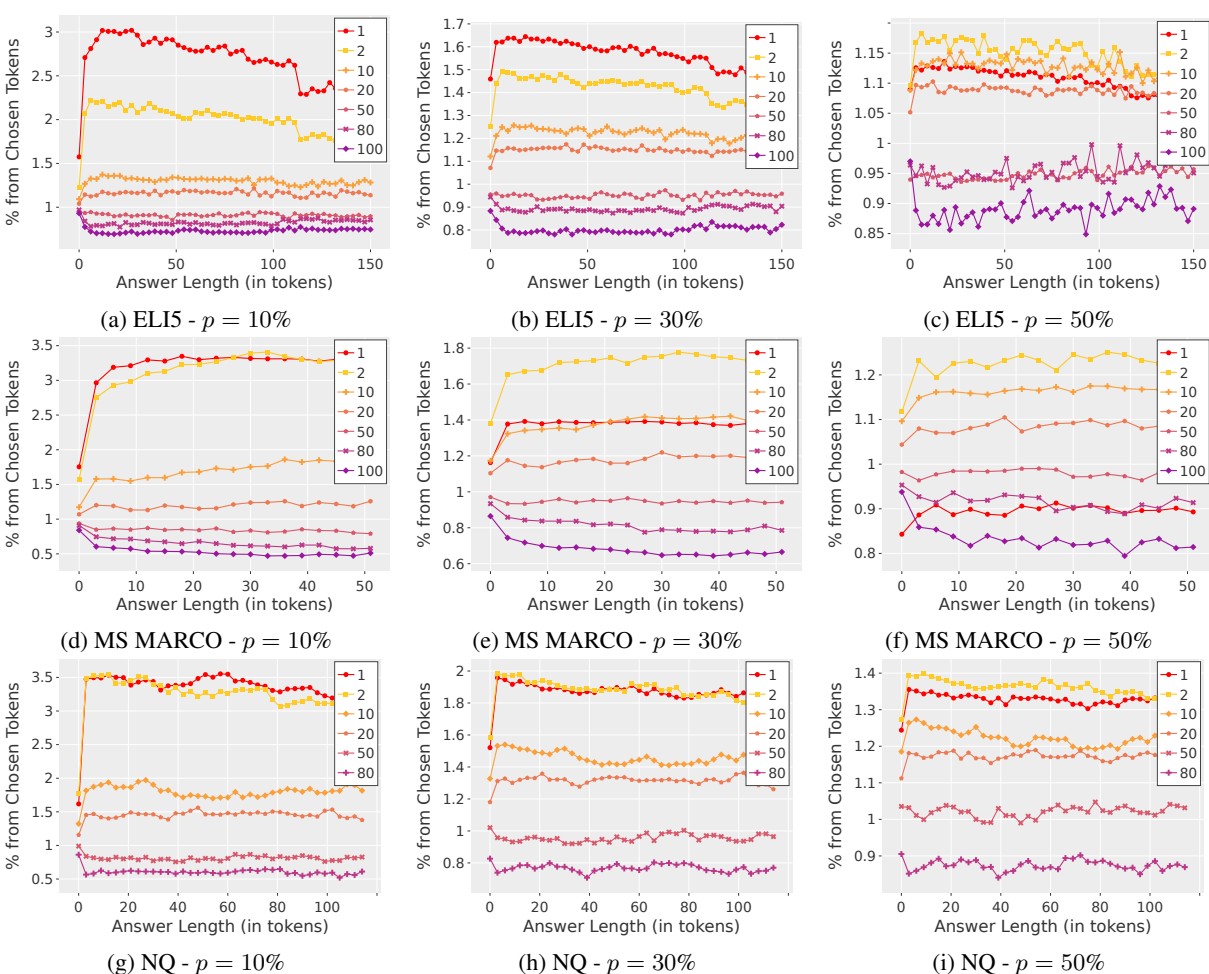

Figure 8: The percentage of tokens that were chosen from each passage, for FiD-Base models. The gold passage (labeled as 1) is colored red. Each row represents a different dataset, and every column represents a different filtering percentage (10,30,50).

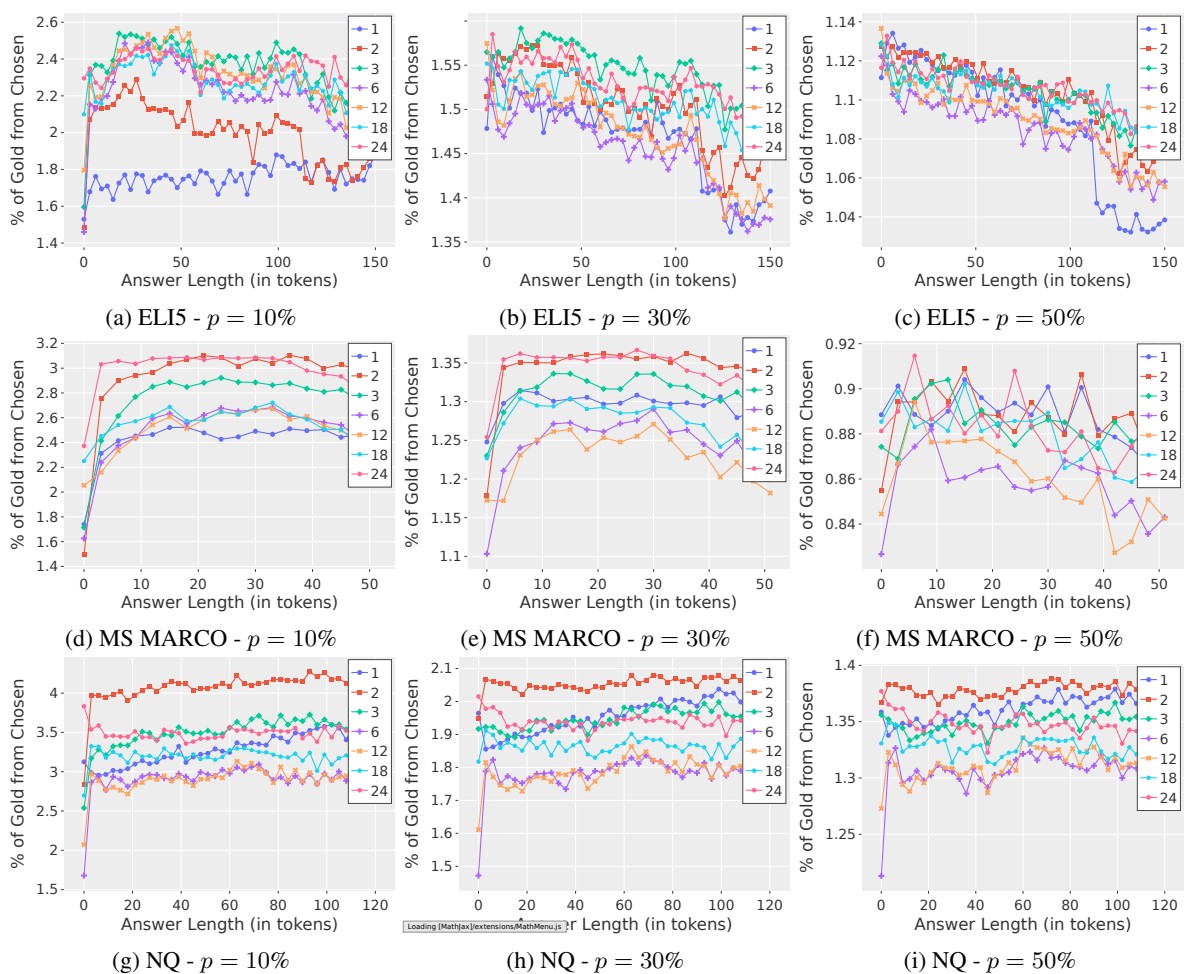

Figure 9: The ratio of tokens that were chosen from the gold passage, per decoder layer (1-24), for FiD-Large models. Each row represents a different dataset, and every column represents a different filtering percentage (10,30,50).

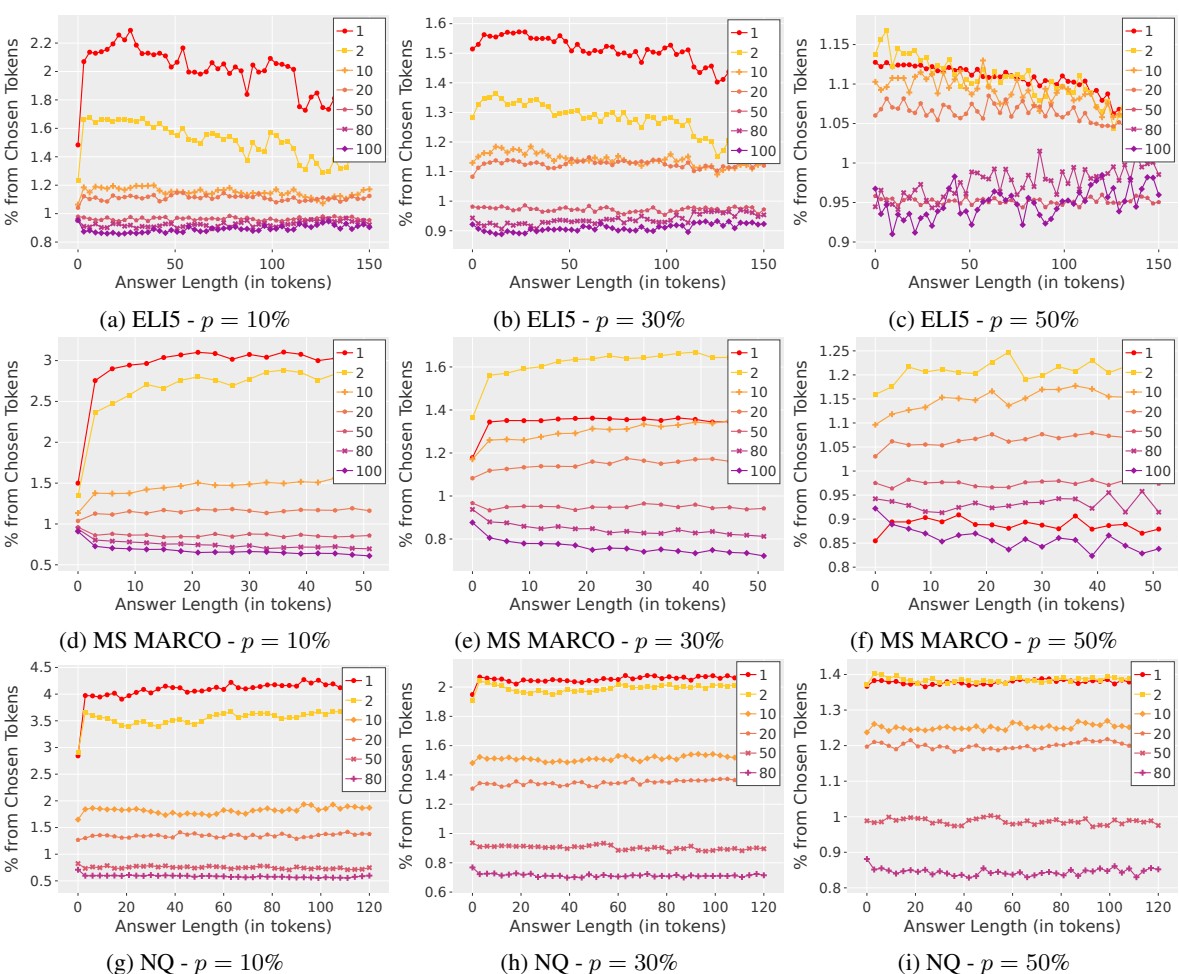

Figure 10: The percentage of tokens that were chosen from each passage, for FiD-Large models. The gold passage (labeled as 1) is colored red. Each row represents a different dataset, and every column represents a different filtering percentage (10,30,50).