# OpenReview forum: "Optimizing Retrieval-augmented Reader Models via Token Elimination"
_EMNLP/2023/Conference — EMNLP 2023 Main_

### Official Review · Reviewer_jDvi · 2023-08-04

**Soundness:** 3

**Excitement:**

3: Ambivalent: It has merits (e.g., it reports state-of-the-art results, the idea is nice), but there are key weaknesses (e.g., it describes incremental work), and it can significantly benefit from another round of revision. However, I won't object to accepting it if my co-reviewers champion it.

**Paper Topic And Main Contributions:**

This paper proposes the token filtering to reduce the time complexity of FiD decoder, keeping only important input tokens which are selected based on the average attention scores, similar to Izacard and Grave (2021a). The proposed token filtering saves substantially the latency and leads to further improved performances, when combining with CALM method.

**Questions For The Authors:**

1) What is the detailed formula of the attention-based score which is used for token filtering?
2) In Table 1-2, what are the results of only applying token filtering, without combining with CALM? What does mean FiD(ours), which are the reproduced results of FiD without applying the token filtering?
3) When the token filtering is done at the decoder layer l, its result affects on filtering tokens at layer l+1? That is, the tokens removed at the previous layer are NOT anymore used after next layers?
4) The token filtering is done per each decoding timestep or per a sequence?
5) Comparing to FiD-light, how much the proposed method saves the latency?
6) Hyperparameters need to be chosen for each latency? Then, in total, because there are 30 distinct latent intervals, we need to keep 30 sets of hyperparameters for enduing a FiD model?
7) Without Max Curve, the methods are not fairly compared? The curves in Figure 5 are those with Max Curve?

**Reasons To Accept:**

The proposed idea of eliminating irrelevant tokens based on the attention score is quite simple, interesting, and well-motivated. In the experiment, the proposed token filtering saves the latency and makes further improvements in terms of generation performance, on various datasets.

**Reasons To Reject:**

- The presentation is not very clearly provided. In particular, the proposed token elimination method in Section 4 is largely unclear. The filtering is performed for each generated token at a specific layer? In Section 5.4, it is also unclear how to control the values of latency; For example, k in the top-k selection is the parameter to control the latency? In the case of the original FiD, how to control the latency in Figure 5?

- Other baseline methods for saving latency need to be compared. For example, instead of using token-level filtering, top k passages can be provided to the FiD, instead of using a full set of input passages. Also, FiD-light could be considered as the baseline for comparing the latency.

**Reproducibility:**

2: Would be hard pressed to reproduce the results. The contribution depends on data that are simply not available outside the author's institution or consortium; not enough details are provided.

**Reviewer Confidence:**

4: Quite sure. I tried to check the important points carefully. It's unlikely, though conceivable, that I missed something that should affect my ratings.

**Typos Grammar Style And Presentation Improvements:**

The method in Section 4 is not very clear. Full details need to be provided for clarity.

---

> ### Author Rebuttal · Authors · 2023-08-28
>
> We appreciate your time and effort in providing constructive feedback, and we address your concern and questions below.
>
>
> **R1**: Reason to reject (1):
>
> We would like to provide a clearer presentation of the token elimination method. The token filtering is performed only once during the decoding process. Before inference, we specify a specific generated token index $t$ and a specific layer $l$. When the decoding process reaches token $t$ and layer $l$, token filtering is done by taking the mean cross-attention scores over input tokens. This creates new filtered cross attention past key values states. Once these key and value states are computed, they are the only ones used for the rest of the decoding process. A more detailed description, including formal equations of the token filtering operation, are provided in the answers to the questions below.
>
> To control the latency of the model (as done in Figure 5), we first run a search over the hyperparameter space on the development set, and get a large amount of combinations, each with a different ROUGE-L and latency value.
>
> For example, we can consider a scenario where we have 200 combination results, with each combination having a latency value ranging from 1 to 5 seconds. Thus, we divide these combinations into, for example, 5 groups, each for a different latency range. According to this number of groups, the first group will have combinations in the range of 1 to 2 seconds, the second one with 2 to 3 seconds, and so on.
>
> For each group, we select the combination with the best ROUGE-L score, ending up with 5 combinations in total, as in Figure 4.
>
> Thus, during inference, we can choose a combination according to the latency of its group. If we wish for low latency, we choose the combination of the first group of 1 to 2 seconds. If we wish for better quality results, while sacrificing latency, we can choose the combination of the group corresponding to 4 to 5 seconds, and so on.
>
> Regarding the top-k selection for FiD, we do indeed perform the suggested operation, and provide FiD with an increasing number of passages (20, 40, 60, 80, 100 passages). We show the results of this input passage amount selection in Figure 5 as red lines, under the name FiD.We will add these exact passage amounts for the standard FiD evaluation for the camera ready version. A more detailed explanation is provided in the answers to questions 6 and 7 below.
>
>
> **R2**: Reason to reject (2):
>
> As mentioned in our response to Reason to reject 1 (R1), we perform the suggested top-k operation, and show the results of this passage amount selection in Figure 5 as red lines, under the name FiD. By using the latency selection method we described in R1, we showcase the other curves, particularly our combined approach with the yellow color. Additionally, we also compare with the baselines of CALM (in green) and Token Filtering (in blue) separately in Figure 5.
>
> Regarding the comparison to FiD-light, we encountered challenges reproducing the FiD-light results, mainly due to the lack of open-sourced code. We note that our approach and FiD-light are complementary and have the potential to be integrated and employed collectively. We elaborate on that more in our answer to question 5, later on in this response.
>
> We plan to open source our code to reproduce our evaluations and datasets upon acceptance.
>
> In addition, we provide answers to your questions below:
>
> **Q1.** We provide the list of operations performed in the token filtering mechanism:
>
> 1. Perform the mean operation over all attention heads.
> 2. Rank the tokens according to their scores.
> 3. Choose the top $p$\% highest ranking tokens.
> 4. Keep only the top tokens in the past cross-attention key value states.
>
> As for a formal definition, the cross-attention scores per input token are defined as follows:
>
> $S_{t,l} = \frac{1}{h} \sum_{i=1}^{h} A_{t,l}^{i}$
>
> where $t$ is the generated token index, $l$ is the layer index, $h$ is the number of attention heads, and $A_{t,l}^{i}$ represents the cross-attention scores at index $t$, layer $l$ and head $i$.
>
> We perform a descending $argsort$ operation on the scores above, and take the top $p$% from the sorted input token indices. Hence, we denote $T$ as the total number of input tokens from all the passages, and $T’$ as the amount of tokens we keep after filtering, which is $p$% from $T$:
>
> $Sorted_{t,l} = argsort(S_{t,l})$
>
> $Top_{t,l} = (Sorted_{t,l} [i])_{i=1}^{T’}$ ,
>
> where $[i]$ indicates accessing the vector at index $i$.
>
> Finally, we keep only the tokens chosen in $Top_{t,l}$ from the cross-attention past key values states $K_{past}, V_{past}$:
>
> $K_{past} = K_{past}[Top_{t,l}], V_{past} = V_{past}[Top_{t,l}]$,
>
> Where $A[B]$ selects the elements from $A$ whose indices appear in $B$.
> These new past key value states are the only ones used for generating all subsequent tokens.
>
> **Q2.** The results for Table-1 for Token Filtering without CALM are presented below, with the reported results from the paper as well:
>
> |          | ELI5  |       | MS MARCO |       |
> |----------|-------|-------|----------|-------|
> |          | R-L   | F1    | R-L      | F1    |
> | FiD Token Filtering | 26.65 | 30.32 | 24.75    | 27.26 |
> | FiD Comb | 26.97 | 31.76 | 25.11 | 27.41 |
>
> As shown above, the results of FiD Comb are better than those of FiD Token Filtering.
>
> Similarly, the results for Table-2 for Token Filtering without CALM on the ELI5 dataset are:
>
> |  | ROUGE-L    |F1   |
> |---------|-------|---------|
> | FiD Token Filtering    | 25.78 | 29.49  |
> | FiD Comb | 25.61 | 29.99 |
>
> Here, we can see that the ROUGE-L is a bit higher, while the F1 score is a bit lower.
>
> FiD (ours) stands for the original FiD model (using T5-Base, and our implementation) and the results shown in Table 1 depict the results of the original FiD model with 100 passages. We will clarify the notation in the camera ready version.
>
> **Q3.** In our implementation, the tokens removed are still kept for layers $l+1,l+2$ and so on. The filtered past key value states will be used from the next generated token.
>
> We performed a preliminary analysis of what happens if the filtering is applied right after layer $l$.
> Below are the results on ELI5:
> We compare our current results from Table 1 in the FiD Comb row, to filtering immediately after $l$, using the same hyperparameter configuration for both:
>
> | Dataset                     | F1    | ROUGE-L | Latency (sec) |
> |-----------------------------|-------|---------|---------------|
> | Filtering Immediately After | 31.88 | 26.9    | 3.653         |
> | FiD Comb                    | 31.76 | 26.97   | 3.833         |
>
> We can observe that the differences between the two are not significant, though we leave further analysis of this comparison for future work.
>
> **Q4.** Per a sequence, the token filtering is performed only once during the decoding process. Before inference, we specify a specific generated token index $t$ and a specific layer $l$. When the decoding process reaches token $t$ and layer $l$, token filtering is performed, thus creating new filtered past key value states. Once these new states are computed, they are the only ones used for the rest of the decoding process.
>
> **Q5.** Although we encountered challenges reproducing the FiD-light results mostly since the lack of open-sourced code, we note that our approach and FiD-light are complementary and have the potential to be integrated and employed collectively. In particular, since FiD-Light utilizes removing input token vectors from the input before the decoder, we can combine their approach with ours. Also, their source-pointing usage can be incorporated as well.
>
> **Q6.** As mentioned in section 5.4 line 376, there are a lot of different hyperparameters to control for (the generated token to filter at, the layer to filter at, etc.), thus requiring a consistent system to choose the best hyperparameter combination for our quality vs. latency tradeoff.
> Thus, we use the division of the combinations into groups, as described in R1 above, which allows for flexibility in choosing the optimal latency setting. Hence, since in the paper we use 30 groups, all the best combinations for every group are kept, but only a single chosen setting is used during inference.
>
> **Q7.** We do require the Max Curve method to compare the methods on the test set in the correct way, which incorporates the quality vs. latency tradeoff. As stated in question 6 and R1 above, each method has many combinations of hyperparameters over the development set. If we apply each one on the test set, and try to compare the methods, it will be difficult to compare them, since each one will be a cluster of points (x-axis is latency, y-axis is ROUGE-L).
>
> Thus, instead of clusters of points, each method is described by a single line in the quality vs. latency plot, composed of the chosen points per group. This allows us to easily compare each method, as shown in Figure 5.
>
> We note that the main focus of our work is on optimizing the quality vs. latency trade-off, hence the focus on the Max Curve method.

---

### Official Review · Reviewer_g1LA · 2023-08-05

**Soundness:** 3

**Excitement:**

3: Ambivalent: It has merits (e.g., it reports state-of-the-art results, the idea is nice), but there are key weaknesses (e.g., it describes incremental work), and it can significantly benefit from another round of revision. However, I won't object to accepting it if my co-reviewers champion it.

**Missing References:**

Some of them might be contemporaneous and just for the authors' references.

[1] Guanghui Qin, Benjamin Van Durme. (2023). Nugget: Neural Agglomerative Embeddings of Text. Proceedings of the 40th International Conference on Machine Learning, in Proceedings of Machine Learning Research 202:28337-28350 Available from https://proceedings.mlr.press/v202/qin23a.html.

[2] David Wingate, Mohammad Shoeybi, and Taylor Sorensen. 2022. Prompt Compression and Contrastive Conditioning for Controllability and Toxicity Reduction in Language Models. In Findings of the Association for Computational Linguistics: EMNLP 2022, pages 5621–5634, Abu Dhabi, United Arab Emirates. Association for Computational Linguistics.

[3] Jesse Mu, Xiang Lisa Li, and Noah Goodman. "Learning to compress prompts with gist tokens." arXiv preprint arXiv:2304.08467 (2023).


**Paper Topic And Main Contributions:**

This paper optimizes retrieval-augmented language models, specifically the FiD model, to reduce decoding time in open domain question answering (ODQA) tasks. It proposes Token Filtering, a token-level elimination method, to remove irrelevant information from retrieved passages. The method reduces runtime by up to 62.2% with minimal performance loss or improvement. Evaluation is done on the Long-Form Question Answering task using the ELI5 dataset.

**Reasons To Accept:**

The paper starts with a bunch of analysis justifying the motivations of filtering unimportant information per layer. The analysis has some insights on how decoders use information, which could benefit the community in future design of decoding algorithms. The proposed method is well-motivated and reasonably backed with experiment results.

**Reasons To Reject:**

1. The paper should have discussed a bit more prior work on token selection/filtering, such as [1], also work on prompt compression[2][3] (some might be contemporaneous, but also have similar work).

2. The proposed idea is a bit incremental since a similar filtering mechanism has been widely adopted in tasks such as coreference resolution, where the difference is the item being filtered, are they tokens or other embeddings (mentions). Therefore, the contribution is more considered as transferring the rank and filter method to apply on token-level (combined with early exiting).

**Reproducibility:**

3: Could reproduce the results with some difficulty. The settings of parameters are underspecified or subjectively determined; the training/evaluation data are not widely available.

**Reviewer Confidence:**

4: Quite sure. I tried to check the important points carefully. It's unlikely, though conceivable, that I missed something that should affect my ratings.

---

> ### Author Rebuttal · Authors · 2023-08-28
>
> We appreciate your time and effort in providing constructive feedback.
>
> Reason to reject (1):
>
> We thank the reviewer for pointing out these relevant references. [1] delves into a learned token filtering component, so it's important to clarify that our method doesn't incorporate any learning. Instead, we analyze the necessary input tokens for direct decoding. Furthermore, our work didn't encompass prompting techniques akin to those in [2] and [3], given the context of questions and passages in our inputs. We will certainly incorporate a comprehensive discussion of these aspects, along with the mentioned references and others, into our paper.
>
> Reason to reject (2):
>
> Ranking and filtering of input tokens constitute a thoroughly explored and significant area of research. As far as we are aware of, we are pioneering the utilization of cross-attention scores for this objective, conducting an in-depth examination of their impacts on extended content generation. While ATLAS (Izacard et al. 2022) did utilize comparable signals, it's worth noting that their focus was on enhancing retriever document filtering, diverging from our approach that centers around reader token filtering.
>
> We plan to open source our code to reproduce our evaluations and datasets upon acceptance.

---

### Official Review · Reviewer_cWny · 2023-08-06

**Soundness:** 4

**Excitement:**

4: Strong: This paper deepens the understanding of some phenomenon or lowers the barriers to an existing research direction.

**Paper Topic And Main Contributions:**

This paper presents an approach how to improve latency without impacting performance for Fusion-in-Decoder (FiD) model. More specifically, the authors tackled the heavy cross-attention computation in the decoder by introducing token filtering and using dynamic decoder layer skipping (aka CALM). They did thorough empirical analysis and showed that decoder serves as the primary source of latency and computational load during inference. They also observed that observe that the decoder’s initial layers (2nd and 3rd) contains majority of the tokens derived from the gold passage.

**Questions For The Authors:**

Question A:
Did you do any qualitative analysis (either using human evaluators or scoring such as BertScore) of what is the impact of your approach in generation output?

**Reasons To Accept:**

Very well written paper and contains through analysis. Improvement in latency is quite significant. Results are also comparable to SOTA.

**Reasons To Reject:**

No major reason to reject.

**Reproducibility:**

4: Could mostly reproduce the results, but there may be some variation because of sample variance or minor variations in their interpretation of the protocol or method.

**Reviewer Confidence:**

3: Pretty sure, but there's a chance I missed something. Although I have a good feel for this area in general, I did not carefully check the paper's details, e.g., the math, experimental design, or novelty.

---

> ### Author Rebuttal · Authors · 2023-08-28
>
> We appreciate your time and effort in providing constructive feedback.
>
> Question A:
>
> While we conducted qualitative assessments (by ourselves) to ensure the coherence and accuracy of the long-form generations, we regrettably couldn't incorporate these findings into the paper due to space constraints. In addition, it is less customary to perform human evaluation for long-form question answering tasks, due to the length of the answers and the complexity of the annotation itself. However, we do intend to address this by including relevant examples upon the paper's acceptance. It's worth mentioning that automatic evaluation metrics like BertScore might not perform effectively on extended generations, as they are primarily designed for sentences or concise paragraphs.
>
>
> We provide the results of the BertScore F1 comparison:
>
> | Model\BertScore F1 | ELI5  | MS MARCO |
> |--------------------|-------|----------|
> | FiD (Ours)         | 83.68 | 85.07    |
> | FiD Comb           | 83.79 | 85.27    |
>
>
> Thus, we observe that our model performs on par and even a bit better overall than the original FiD implementation.

---

### Meta-Review · Area_Chair_yK3Z · 2023-09-05

**Recommendation:** 5
**Best Paper Recommendation:** No

**Metareview:**

The paper proposes a method of token filtering for reducing latency when using the Fusion-in-Decoder model for retrieval-based tasks (e.g. open domain QA). All reviewers see the problem as well-motivated and the method as useful, with good experimental justification and analysis. R2 notes that the method may be somewhat incremental but is mostly favorable towards it. R3 is mainly concerned with the presentation, and thinks details in the paper could be better clarified. Additionally, they think more baselines could be added. The authors have responded to this by more formally defining the method proposed in the paper and clarifying various aspects e.g. how the extensive number of hyperparameters is selected, how token filtering is performed, etc.. The overall perception is that the paper is sound.

**Meta-Review:**

The paper proposes a method of token filtering for reducing latency when using the Fusion-in-Decoder model for retrieval-based tasks (e.g. open domain QA). All reviewers see the problem as well-motivated and the method as useful, with good experimental justification and analysis. R2 notes that the method may be somewhat incremental but is mostly favorable towards it. R3 is mainly concerned with the presentation, and thinks details in the paper could be better clarified. Additionally, they think more baselines could be added. The authors have responded to this by more formally defining the method proposed in the paper and clarifying various aspects e.g. how the extensive number of hyperparameters is selected, how token filtering is performed, etc.. The overall perception is that the paper is sound.

---

### Decision · Program_Chairs · 2023-10-07

**Decision:**

Accept-Main

**Comment:**

The paper proposes a method of token filtering for reducing latency when using the Fusion-in-Decoder model for retrieval-based tasks (e.g. open domain QA). All reviewers see the problem as well-motivated and the method as useful, with good experimental justification and analysis. R2 notes that the method may be somewhat incremental but is mostly favorable towards it. R3 is mainly concerned with the presentation, and thinks details in the paper could be better clarified. Additionally, they think more baselines could be added. The authors have responded to this by more formally defining the method proposed in the paper and clarifying various aspects e.g. how the extensive number of hyperparameters is selected, how token filtering is performed, etc.. The overall perception is that the paper is sound.